# Effects of Thymol and Carvacrol Eutectic on Growth Performance, Serum Biochemical Parameters, and Intestinal Health in Broiler Chickens

**DOI:** 10.3390/ani13132242

**Published:** 2023-07-07

**Authors:** Lixuan Li, Xiaochun Chen, Keying Zhang, Gang Tian, Xuemei Ding, Shiping Bai, Qiufeng Zeng

**Affiliations:** 1Key Laboratory of Animal Disease-Resistance Nutrition, Ministry of Education, Ministry of Agriculture and Rural Affairs, Key Laboratory of Sichuan Province, Animal Nutrition Institute, Sichuan Agricultural University, Chengdu 611130, China; rosalind20202021@163.com (L.L.); zkeying@sicau.edu.cn (K.Z.); tgang2008@163.com (G.T.); dingxuemei0306@163.com (X.D.); shipingbai@sicau.edu.cn (S.B.); 2Institute of Animal Science, Chengdu Agricultural College, Chengdu 611130, China; chenxiaochun163@163.com

**Keywords:** broilers, thymol and carvacrol eutectic, serum immunoglobulins, intestinal morphology and function

## Abstract

**Simple Summary:**

With the demand for high-quality poultry products, it is imperative to exploit effective and green feed additives that can stimulate the latent productive capacity of broiler chickens. Numerous studies have shown that essential oils (EOs) are able to maintain intestinal microflora stabilization, improve the digestion and absorption efficiency of intestine, and have effective anti-inflammatory and antioxidant properties. However, essential oils themselves have many limitations, for example, poor stability and palatability. In this study, we evaluated thymol and carvacrol eutectic (TCE), which was produced using pharmaceutical cocrystal technology in broilers’ feed. The results showed that supplementation with TCE in broiler diets can improve growth performance, jejunal morphology, and nutrient absorption and transport capacities, as well as ileal morphology, barrier function, and inflammatory status, which indicates that the bioactive compounds of essential oils treated using pharmaceutical cocrystal technology can play an active role in poultry production.

**Abstract:**

This study aimed to evaluate the effect of diets supplementing with various levels of thymol and carvacrol eutectic (TCE) on growth performance, serum biochemical parameters, intestinal morphology, and the expression of intestinal nutrient absorption, barrier function- and inflammation-related genes in broiler chickens. A total of 640 one-day-old Arbor Acres male broilers with similar body weights were randomly divided into four groups (8 replicates/group, 20 broilers/replicate). Birds in the four experimental groups were fed a basal diet with TCE at 0, 30, 60, or 120 mg/kg. The results showed that the growth performance of birds during 22–42 d or 1–42 d, serum IgE and IgG content at 21 d of age, jejunal and ileal morphology, ileal *MUC2*, *OCLN*, and *IL-10* mRNA expression were significantly increased compared with the control group (*p* < 0.05), and the ileal *IL-6* mRNA expression quadratically decreased (*p* < 0.05) with increasing dietary TCE supplemented dosage, and its expression showed a linear downward trend (0.05 < *p* < 0.1). Meanwhile, compared with the other three groups, birds fed diets with 30 mg/kg TCE presented better (*p* < 0.05) growth performance, intestinal morphology, and function. These results indicated that the optimal supplementation amount of TCE in the broiler diets was 30 mg/kg.

## 1. Introduction

With the demand for high-quality poultry products, it is imperative to exploit effective and green feed additives that can stimulate the latent productive capacity of broiler chickens. Thus, plant essential oils (EOs) have gradually become a research hotspot in the animal feed industry. Recently, many studies have verified that EOs have anti-inflammatory, antibacterial, antioxidant, immunoregulatory, and other characteristics [1,2,3] and have the potential to enhance disease resistance, enhance production performance, and improve intestinal health [4,5,6]. Gholami-Ahangaran et al. (2022) found that thymol (an essential ingredient of EOs) can maintain intestinal microflora stabilization and has effective anti-inflammatory and antioxidant properties, ultimately improving the performance of animals [7]. Liu et al. (2019) observed that broilers orally administered essential oils with the active ingredient carvacrol could inhibit the secretion of inflammatory cytokines caused by *lipopolysaccharide* and showed an anti-inflammatory function [8].

Plant EOs are volatile, natural, and complex mixtures of compounds that are mainly extracted from whole grass, fruits, leaves, flowers, roots, skins, bark, gum, etc. [9]. The major components of many EOs are phenolic compounds (terpenoids and phenylpropanoids), such as thymol, carvacrol, and eugenol. Therefore, EOs themselves have many limitations. For example, EOs have a strong smell and poor palatability and are unstable and vulnerable to changes caused by external factors [10]. The stability and biological activity of EOs may be affected by temperature, light, metal, water, and oxygen in the production system, which may lead to their deterioration [11]. Therefore, it is important to find new methods to improve the stability and palatability of EOs.

Recently, with the development of interdisciplinary research, the use of pharmaceutical cocrystal technology in the field of medicine has attracted great attention. Pharmaceutical cocrystal technology can allow the controllable construction of an active pharmaceutical ingredient (API) in the same crystal lattice with other ligands in hydrogen bonding, π–π stacking, van der Waals, and other noncovalent linkages [12,13], which improves the physicochemical properties of the API without altering its structure and biological activity, including increased solubility and dissolution rate and improved chemical stability, mechanical properties, and bioavailability [14]. Moreover, the two isomers, thymol and carvacrol, among thousands of EOs constituents, possess high antibacterial activity and are also major components of commonly used herbs such as thyme and oregano [15]. Du et al. (2016) found that dietary supplementation with 60, 120, and 240 mg/kg EOs (which contained 25% thymol and 25% carvacrol as active components) alleviated intestinal lesions, improved intestinal histomorphology, decreased the inflammatory response, and enhanced the specific immune response in *C. perfringens*-challenged broiler chickens [16]. Thus, it is interesting and meaningful to evaluate the effectiveness of applying thymol and carvacrol eutectic (TCE) in broilers, which can provide new insight for the development of EOs products. The objective of this study, therefore, was to evaluate the effect of dietary supplementation with different dosages of TCE on the growth performance, intestinal digestive and absorptive capacities, and intestinal health in broiler chickens and then to recommend the optimal supplementation amount of TCE based on linear and quadratic regression analysis.

## 2. Materials and Methods

The Institutional Animal Care and Use Committee (IACUC) of Sichuan Agricultural University approved all procedures used in the study.

### 2.1. Birds, Experimental Design, Diet, and Management

A total of 640 one-day-old male healthy Arbor Acres broiler chicks with an average weight of 43.06 ± 0.19 g were randomly allocated to 4 experimental groups. Each group had 8 replicates, and each replicate contained 20 birds. Birds in the 4 experimental groups were fed a basal diet with TCE at 0, 30, 60, or 120 mg/kg. The TCE was a 1:1 eutectic of thymol and carvacrol, which was made and provided by Shanghai Institute of Materia Medica, Chinese Academy of Science, China. The composition and nutrient levels of the basal diet are shown in Table 1. The diet was offered in pelleted form. All broilers were housed in cages (length 100 × width 80 × height 60 cm) in a humidity- and temperature-controlled room and had free access to water and feed. At 1–3 days of age, the initial temperature of broilers pens was maintained at 32–35 °C and then gradually decreased until reaching 22 °C, and relative humidity was maintained at 60–70% during 1–10 days of age and then gradually decreased to about 55%.

### 2.2. Data and Sample Collection

At 21 and 42 days of age, after 12 h of feed withdrawal, birds were weighed, and the feed intake was obtained by cages. Body weight (BW), average daily gain (ADG), average daily feed intake (ADFI), and feed-to-gain ratio (F:G) were calculated for the periods of 1–21 d, 22–42 d, and 1–42 d of age.

Then, one of them with an average weight was selected from each replicate for blood collection (n = 8). Blood was collected via the vena brachialis and placed into tubes without heparin sodium (an anticoagulant), and then, serum was isolated and stored at −20 °C for serum biochemical determination. The birds from which blood was collected were euthanized via exsanguination. The jejunum and ileum samples were fixed in a solution of 4% paraformaldehyde to determine intestinal morphology. The remaining ileum segments were rinsed with normal saline, the contents were removed, and mucous membranes were gently scraped and rapidly frozen in liquid nitrogen and then stored at −80 °C for RT-qPCR.

### 2.3. Serum Biochemical Indices

Serum total protein (TP), albumin (ALB), globulin (GLB), and uric acid levels were determined using an automated blood chemistry analyzer (Hitachi-3100; Hitachi Medical, Tokyo, Japan). Serum immunoglobulin G (IgG), immunoglobulin M (IgM), and immunoglobulin E (IgE) levels were measured via enzyme-linked immunosorbent assay (ELISA) on a Rayto RT-6100 microplate reader. The assay kits were supplied by Nanjing Jiancheng Bioengineering Institute of China.

### 2.4. Intestinal Morphology

Briefly, the intestinal segments fixed with 4% paraformaldehyde were washed and then subjected to a series of procedures according to our previous study by Han et al. (2017) to measure villus height (VH), crypt depth (CD), villus-to-crypt ratio (VH:CD) and intestinal wall thickness (IWT) [17].

### 2.5. Gene Expression

The method of RNA extraction and the procession of reverse transcription and real-time PCR were performed according to previously reported methods [18]. Briefly, total RNA was extracted from the mucosa of the jejunum and ileum using RNAiso Plus reagent (Takara, Dalian, China). The integrity, concentration, and purity of RNA were determined using a NanoDrop ND-2000 spectrophotometer (Thermo Scientific, Wilmington, DE, USA). A SuperScript™ II Reverse Transcriptase kit (Invitrogen, Carlsbad, CA, USA) was used to synthesize cDNA. The cDNA was subsequently diluted to 10 ng/µL for qRT-PCR analysis.

Gene expression was finally normalized to *β-actin*, and the relative expression level of each gene was calculated using the 2^–ΔΔCt^ method. The primers for analysis of gene expression of peptide transporter 1 (*PepT1*), cationic amino acid transporter 1 (*CAT-1*), and fatty acid transport protein 4 (*FATP4*) in the jejunal mucosa, as well as occludin (*OCLN*), zonula occludens-1 (*ZO-1*), mucin 2 (*MUC2*), tumor necrosis factor-α (*TNF-α*), interleukin-10 (*IL-10*), interleukin- 1β (*IL-1β*), and interleukin-6 (*IL-6*) in the ileal mucosa, are shown in Table 2.

### 2.6. Statistical Analysis

The effects of the supplemented dosage of TCE in diets among the 4 treatments were analyzed via one-way ANOVA using the GLM procedure in SAS software (SAS Institute Inc., Cary, NC, USA). When the effect of dietary treatment was significant (*p* < 0.05), means were compared using the LSD procedure in SAS software. Meanwhile, when the effect of dietary treatments was significant (*p* < 0.05), polynomial contrasts and the linearity of response to dietary TCE supplemented amount were examined using linear and quadratic regression. Probability values ≤ 0.05 were considered significant.

## 3. Results

### 3.1. Growth Performance

Table 3 presented that feeding with diets supplemented with different dosages of TCE has no significant effects (*p* > 0.05) on the growth performance of broilers at 1–21 d of age. However, the BW at 42 d, the ADG from 22 to 42 d or from 1 to 42 d linearly and quadratically increased (*p* < 0.05), and the F:G from 22 to 42 d or from 1 to 42 d linearly decreased (*p* < 0.05) with increasing dietary TCE supplementation dosage.

### 3.2. Serum Biochemical Indices

As shown in Table 4, TCE supplementation in diets had no significant effect (*p* > 0.05) on the serum TP, ALB, GLB, uric acid, IgE (d 42), IgG (d 42), and IgM concentrations of broilers. However, the serum IgE and IgG content at 21 d of age presented a linear increase (*p* < 0.05) with the increase in dietary TCE-supplemented dosage.

### 3.3. Intestinal Morphology

As shown in Table 5, supplementation with different dosages of TCE in diets had a linear or quadratic effect (*p* < 0.05) on jejunal VH (d 42), CD, VH:CD (d 21), and IWT (d42) as well as on ileal VH (d 21), CD (d 42), VH:CD (d 21) and IWT (d 21). Compared with the other three groups, TCE supplemented at 30 mg/kg significantly increased (*p* < 0.05) jejunal VH (d 21 and d 42) and IWT (d 42) and significantly decreased (*p* < 0.05) ileal CD (d 21). Overall, broilers fed diets supplemented with TCE at 30 mg/kg presented a better jejunal morphology, and those fed diets supplemented with 30 and 60 mg/kg showed a better ileal morphology.

### 3.4. Intestinal Nutrient Absorption, Barrier Function, and Inflammation-Related Gene Expression

As shown in Figure 1a–c, with the increase in dietary TCE supplementation dosage, jejunal gene expression of *CAT-1* (d 42) presented a quadratic change that first decreased and then increased (*p* < 0.05).

As shown in Figure 1d–f, the mRNA expression of ileal barrier function-related genes, such as *MUC2* (42 d), presented a linear or quadratic increase (*p* < 0.05), and *OCLN* (42 d) or *ZO-1* (21d) expression showed a quadratic decrease or increase (*p* < 0.05), with increasing dietary TCE supplemented dosage.

As shown in Figure 1g–j, the mRNA abundance of ileal inflammatory *IL-6* showed a quadratic decrease at 42 d (*p* < 0.05), but ileal anti-inflammatory *IL-10* (42d) showed a linear or quadratic increase (*p* < 0.05) with increasing dietary TCE supplementation dosage. Compared with the other three groups, TCE supplemented at 30 mg/kg significantly decreased (*p* < 0.05) ileal *TNF-α* mRNA expression in 21 d.

## 4. Discussion

In this present study, we comprehensively evaluated the effect of TCE on growth performance, serum biochemical parameters, and immunoglobulin content, as well as intestinal morphology and function, in broiler chickens. As with many previous studies, the results in our current study also showed that dietary supplementation with various levels of TCE had a significant positive effect on broiler growth performance, especially at the supplementation dosage of 30 mg/kg. Ruan et al. (2021) found that the addition of 150 or 300 mg/kg oregano essential oil to the diet significantly increased the ADFI and ADG of yellow-feathered broilers [19]. Youssef et al. (2021) observed that supplementation with a blend of octagon, rosemary, and thyme oregano essential oils at 25 mg/kg in broilers at 22–42 days of age significantly increased body weight gain [20]. Zhang et al. (2021) also found that dietary supplementation with either natural or synthetic oregano essential oil increased ADG and decreased F:G in broilers at 1–21 days of age compared with the control group [21]. The above studies all showed a positive effect of EOs, but the optimal supplementation dose of EOs was different, which may be due to the effective ingredients of EOs from different sources or EOs product forms being different.

Analysis of blood parameters is helpful in assessing the general health of birds. In our study, the results showed that supplementation with various dosages of TCE had no effect on serum total protein, albumin, globulin, and uric acid concentrations in broilers. The reason may be that blood proteins mainly reflect the indicators of the animal’s physical health and nutritional status and maintenance of homeostasis [22], and no differences emerged to illustrate that the addition of TCE did not affect the health and nutritional status of broilers. Moreover, immunoglobulins are closely related to immune function, IgG can actively participate in humoral immunity [23]. Our present study found that the serum IgE and IgG concentrations of broilers at 21 d of age linearly increased with increasing TCE supplementation levels. Mohiti-Asli and Ghanaatparast-Rashti (2017) also found that broilers fed 300 mg/kg oregano essential oil in the diet had higher IgG titers and total antibody titers than broilers fed the control diet [24].

The growth performance of broilers is closely related to the health of the gastrointestinal tract, including intestinal morphology, digestion and absorption efficiency of nutrients, intestinal barrier function, and so on [25]. It has been reported that the intestinal VH, CD, and VH/CD are important measures of the digestive and absorptive function of the small intestine [26,27]. In our study, we found that broilers fed diets supplemented with TCE at 30 mg/kg presented a better jejunal morphology, and those supplemented with TCE at 30 and 60 mg/kg showed a better ileal morphology. The changes in jejunal or ileal morphology were consistent with the production performance in this present study. Similarly, Ding et al. (2020) found that the ileal VH and V/C ratio of the ileum and jejunum of broilers in the essential-oil-supplemented group were highest, and the CD of the duodenum was lowest compared with the control group [28].

Regarding the health status of the animal intestine, the intestinal mucosal barrier and inflammatory status can be examined. Intestinal tight junction proteins play an important role in intestinal epithelial barrier function [29,30]. In this current study, we found that *MUC2* and *OCLN* mRNA expression levels were upregulated in the ileum of broilers fed diets supplemented with TCE. Similarly, Wang et al. (2019) reported that compound EOs with thymol as an active ingredient quadratically increased *MUC2* mRNA expression in the ileum of laying hens at week 25 [31]. Dietary supplementation of EOs with 25% thymol and 25% carvacrol as active components tended to linearly increase the expression of the *OCLN* gene [16]. In addition, inflammatory cytokines play an important role in the immune response and inflammation and are also important mediators of intestinal dysfunction [32]. In this study, *IL-10* mRNA expression linearly increased, and *IL-6* mRNA expression linearly decreased with the addition of TCE, and the expression level of *TNF-α* was significantly reduced on the 21th d after the addition of 30 mg/kg TCE. It has been reported that the anti-inflammatory cytokine *IL-10* is particularly important in maintaining intestinal microbial immune homeostasis and can suppress immune responses to foreign antigens [33]. The cytokine *IL-10* mainly inhibits the release of proinflammatory mediators, including *TNF-α*, *IL-1β*, and *IL-6*, to target monocytes and macrophages [34]. Liu et al. (2019) also found that oral administration of carvol essential oil to broilers challenged by *lipopolysaccharide* inhibited the gene expression of *TNF-α*, *IL-1β*, and *IL-6* [8]. These results indicated that similar to other EOs products, TCE could improve intestinal function.

## 5. Conclusions

In conclusion, supplementation with TCE in broiler diets can improve growth performance and jejunal morphology, nutrient absorption and transport capacities, as well as ileal morphology, barrier function, and inflammatory status. The optimal supplemented dosage of TCE in diets for broilers was 30 mg/kg. These results suggest that pharmaceutical cocrystal technology can be used to produce a new form of EOs. However, more concerns are needed about the applied advantages of EOs Eutectic in the poultry feed industry.

## Figures and Tables

**Figure 1 animals-13-02242-f001:**
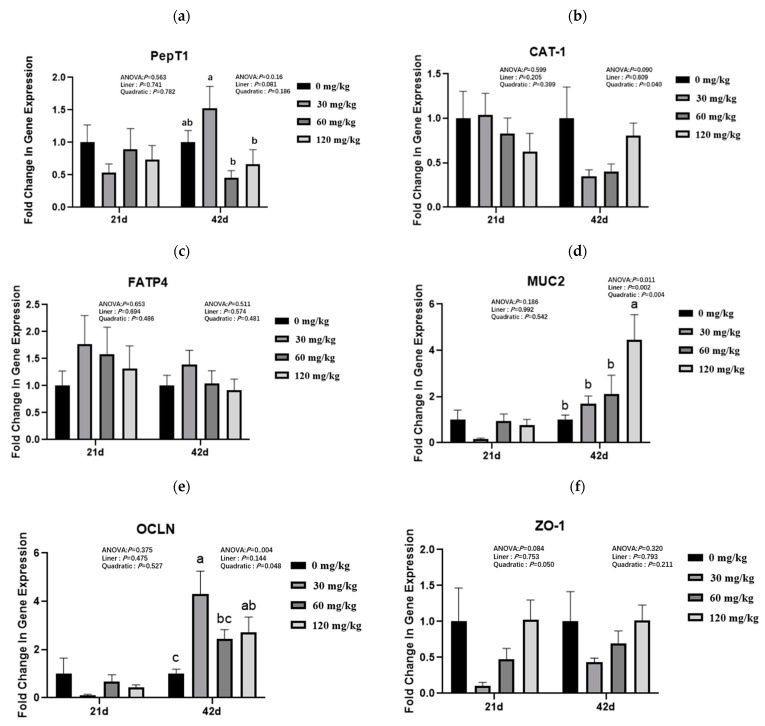
Effects of thymol and carvacrol eutectic on jejunal nutrient transport genes expression of broilers. Values are the means of 8 chickens per dietary treatment. Error bars represent SEM. The related mRNA expression of (**a**) *PepT1*, peptide transporter 1, (**b**) *CAT1*, cationic amino acid transporter 1, and (**c**) *FATP4*, fatty acid transport protein 4 of jejunum in broilers on d 21 and d 42 via the RT-PCR; (**d**) *MUC2*, mucin 2, (**e**) *OCLN*, occludin, and (**f**) *ZO-1*, zonula occludens-1 of ileum in broilers on d 21 and d 48 via the RT-PCR; (**g**) *IL-6*, interleukin-6, (**h**) *IL-1β*, interleukin- 1β, (**i**) *TNF-α*, tumor necrosis factor-α, (**j**) *IL-10*, interleukin-10 of broilers on d 21 and 48 via the RT-PCR. a–c Mean value above each bar with no common superscript differs significantly (*p* < 0.05).

**Table 1 animals-13-02242-t001:** The composition and nutrition levels of the basal diet (air-dry basis).

Ingredients	Contents (%)	
	1–21 d	22–42 d
Corn	50.42	50.41
Flour	4.00	4.00
Soybean meal	37.9	36.0
Soya oil	3.65	5.33
Dicalcium phosphate	1.90	1.30
Limestone	1.26	1.33
Sodium chloride	0.30	0.30
Vitamin premix ^(1)^	0.03	0.03
Mineral premix ^(2)^	0.20	0.20
L-Lysine-HCL	0.00	0.07
L-Threonine	0.00	0.03
DL-Methionine	0.19	0.21
Choline chloride (50%)	0.15	0.15
Rice bran	0.00	0.64
Total	100.00	100.00
Calculated nutritional levels, %		
ME, Mcal/kg	3.00	3.10
Crude protein	20.88	20.16
Calcium	1.00	0.90
Non-phytate phosphorus	0.45	0.35
Digestible lysine	1.15	1.15
Digestible methionine	0.50	0.50
Digestible threonine	0.81	0.81

^(1)^ Vitamin premix provided the following per kilogram of final diets: vitamin A 7200 IU; vitamin D_3_ 1440 IU; vitamin E 60 IU; vitamin K_3_ 2.88 mg; vitamin B_1_ 1.20 mg; vitamin B_2_ 4.32 mg; vitamin B_6_ 2.16 mg; vitamin B_12_ 0.02 mg; biotin 0.29 mg; folic acid 2.40 mg; nicotinamide 24.00 mg; pantothenic acid 15.00 mg. ^(2)^ Mineral premix provided the following per kilogram of final diets: copper (CuSO_4_·5H_2_O) 10.0 mg; iron (FeSO_4_·H_2_O) 80.00 mg; zinc (ZnSO_4_·H_2_O) 60.00 mg; manganese (MnSO_4_·H_2_O) 100.00 mg; iodine (KI) 0.45mg; selenium (Na_2_SeO_3_) 0.30 mg.

**Table 2 animals-13-02242-t002:** Primer pairs used for RT-qPCR analyses ^1^.

Gene Name/Abbreviation	Accession Number	Primer Sequence (5′-3′)
*β-actin*	NM_205518	F:GAGAAATTGTGCGTGACATCA
		R:CCTGAACCTCTCATTGCCA
*FATP4*	XM_015279553.3	F:GAGCCGCATCCTCAACCTG
		R:GCTGCCATTCCTGCCTTCC
*PepT1*	NM_204365.2	F:TCACTGTTGGCATGTTCCT
		R:TTCGCATTGCTATCACCTA
*CAT-1*	NM_001145490.1	F:TACAACAGGTGAGGAGGTG
		R:AAGCCACAAAGCAGATGAG
*ZO-1*	XM_040680632.1	F:GGCAAGTTGAAGATGGTGGT
		R:ATGCCAGCGACTGAATTTCT
*MUC2*	XM_040701667.1	F:AACTCCTCCTTTGTATGCG
		R:ATTCAACCTTCTGCCCTAA
*OCLN*	NM_205128.1	F:CCTCATCGTCATCCTGCTCTG
		R:GCCACGTTCTTCACCCACTC
*IL-10*	NM_001004414.2	F:TGTCACCGCTTCTTCACCT
		R:TCCCGTTCTCATCCATCTT
*IL-6*	NM_204628.1	F: CTCCTCGCCAATCTGAAGTC
		R:CCCTCACGGTCTTCTCCATA
*IL-1* *β*	NM_204524.1	F:GCTCTACATGTCGTGTGTGATGAG
		R:TGTCGATGTCCCGCATGA
*TNF-α*	NM_204267.2	F:TACCCTGTCCCACAACCTG
		R:GGCGGTCATAGAACAGCAC

^1^ F, forward; R, reverse; *FATP4*, fatty acid transport protein 4; *PepT1*, peptide transporter 1; *CAT-1*, cationic amino acid transporter 1; *ZO-1*, zonula occludens-1; *MUC2*, mucin 2; *OCLN*, Occludin; *IL-10*, interleukin-10; *IL-6*, interleukin-6; *IL-1β*, interleukin-1β; *TNF-α*, tumor necrosis factor-α.

**Table 3 animals-13-02242-t003:** Effects of thymol and carvacrol eutectic on growth performance of broilers ^1^.

Item	The Supplemented Dosage of Eutectic, mg/kg				SEM	*p*-Value		
	0	30	60	120		ANOVA	Linear	Quadratic
Body weight (BW), g								
1 d	43.05	43.15	43.08	42.98	0.09	0.327	0.321	0.197
21 d	914.5	933.3	935.7	927.4	15.86	0.552	0.415	0.345
42 d	2704^b^	2866 ^a^	2809 ^a^	2856 ^a^	49.56	0.011	0.026	0.027
Average daily gain (ADG), g								
1–21 d	41.50	42.39	42.50	42.11	0.76	0.554	0.412	0.348
22–42 d	85.27 ^b^	92.13 ^a^	89.25 ^b^	91.93 ^a^	1.95	0.005	0.018	0.025
1–42 d	63.37 ^b^	67.23 ^a^	65.86 ^a^	66.98 ^a^	1.18	0.011	0.026	0.027
Average daily feed intake (ADFI), g								
1–21 d	67.55	67.05	68.05	67.22	1.53	0.917	0.997	0.988
22–42 d	157.5	166.4	162.5	163.1	3.53	0.114	0.280	0.159
1–42 d	116.5	120.7	119.4	119.2	2.20	0.309	0.354	0.252
Feed to gain ratio (F: G), g/g								
1–21 d	1.63	1.58	1.60	1.60	0.03	0.447	0.420	0.432
22–42 d	1.85	1.81	1.82	1.78	0.03	0.052	0.017	0.060
1–42 d	1.80	1.76	1.78	1.75	0.02	0.054	0.025	0.078

^a, b^ Values within a column with no common superscripts differ significantly (*p* < 0.05). ^1^ Values are the means of 8 replicates of 20 chickens each.

**Table 4 animals-13-02242-t004:** Effects of thymol and carvacrol eutectic on serum biochemical indexes of broilers ^1^.

Item	The Supplemented Dosage of Eutectic, mg/kg				SEM	*p*-Value		
	0	30	60	120		ANOVA	Linear	Quadratic
Serum biochemical indexes of broilers at 21 d of age								
Total protein (g/L)	26.20	24.73	27.01	25.49	1.20	0.281	0.970	0.999
Albumin (g/L)	12.20	11.53	12.36	11.59	0.72	0.566	0.664	0.907
Globulin (g/L)	14.00	13.20	14.65	13.90	0.87	0.440	0.682	0.920
Uric Acid (μmol/L)	227.6	288.9	268.5	300.3	37.28	0.244	0.102	0.230
IgE (U/mL)	6.96	8.59	9.80	10.09	1.28	0.082	0.012	0.033
IgG (mg/mL)	17.10	17.92	21.25	23.99	3.63	0.230	0.039	0.116
IgM (mg/mL)	9.18	6.54	8.26	9.83	1.63	0.225	0.497	0.161
Serum biochemical indexes of broilers at 42 d of age								
Total protein (g/L)	27.26	27.56	25.13	23.29	2.35	0.248	0.057	0.136
Albumin (g/L)	12.40	13.29	11.83	10.71	1.14	0.172	0.081	0.105
Globulin(g/L)	14.86	14.28	13.30	12.40	1.40	0.330	0.061	0.175
Uric Acid (μmol/L)	202.1	247.5	182.0	191.5	43.20	0.453	0.482	0.664
IgE (U/mL)	4.96	5.79	4.93	5.74	1.12	0.781	0.958	0.990
IgG (mg/mL)	45.24	51.08	48.41	58.01	5.67	0.163	0.095	0.227
IgM (mg/mL)	10.13	8.11	3.95	11.61	4.39	0.347	0.871	0.315

^1^ Values are the means of 8 chickens per dietary treatment.

**Table 5 animals-13-02242-t005:** Effects of thymol and carvacrol eutectic on intestinal morphology of broilers ^1^.

Item	The Supplemented Dosage of Eutectic, mg/kg				SEM	*p*-Value		
	0	30	60	120		ANOVA	Linear	Quadratic
Jejunal morphology								
Villus height at 21 d (μm)	1003 ^b^	1099 ^a^	997.1 ^b^	1036 ^b^	32.26	0.006	0.882	0.499
Crypt depth at 21 d (μm)	192.2 ^b^	197.6 ^b^	233.2 ^a^	232.2 ^a^	9.09	<0.001	<0.001	<0.001
VH:CD at 21 d	5.41 ^a^	5.77 ^a^	4.58 ^b^	4.77 ^b^	0.22	<0.001	<0.001	<0.001
Intestinal wall thickness at 21 d (μm)	1374	1462	1396	1434	39.76	0.128	0.413	0.498
Villus height at 42 d (μm)	1147 ^b^	1460 ^a^	1161 ^b^	1119 ^b^	52.74	<0.001	0.035	<0.001
Crypt depth at 42 d (μm)	207.5 ^a^	216.7 ^a^	185.0 ^b^	188.0 ^b^	9.29	0.002	0.003	0.011
VH:CD at 42 d	6.16 ^b^	7.36 ^a^	7.06 ^a^	6.03 ^b^	0.43	0.004	0.603	0.001
Intestinal wall thickness at 42 d (μm)	1563 ^b^	1993 ^a^	1613 ^b^	1539 ^b^	55.65	<0.001	0.024	<0.001
Ileal morphology								
Villus height at 21 d (μm)	772.5 ^a^	654.7 ^c^	704.3 ^b^	601.8 ^d^	19.64	<0.001	<0.001	<0.001
Crypt depth at 21 d (μm)	245.4 ^a^	205.1 ^c^	249.0 ^a^	221.5 ^b^	6.64	<0.001	0.225	0.217
VH:CD at 21 d	3.23 ^a^	3.30 ^a^	2.86 ^b^	2.72 ^b^	0.09	<0.001	<0.001	<0.001
Intestinal wall thickness at 21 d (μm)	1272 ^a^	1046 ^c^	1177 ^b^	1016 ^c^	30.12	<0.001	<0.001	<0.001
Villus height at 42 d (μm)	1086 ^b^	1202 ^a^	1226 ^a^	996.5 ^c^	39.95	<0.001	0.193	<0.001
Crypt depth at 42 d (μm)	199.9 ^ab^	205.5 ^a^	191.6 ^b^	175.0 ^c^	6.69	<0.001	<0.001	<0.001
VH:CD at 42 d	5.56 ^b^	5.89 ^b^	6.73 ^a^	5.65 ^b^	0.22	<0.001	0.062	<0.001
Intestinal wall thickness at 42 d (μm)	1644 ^b^	1861 ^a^	1846 ^a^	1559 ^b^	52.61	<0.001	0.305	<0.001

^a–d^ Values within a column with no common superscripts differ significantly (*p* < 0.05). ^1^ Values are the means of 8 chickens per dietary treatment.

## Data Availability

Data is available upon request from the corresponding authors.

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
