# Peer review of "Effects of Thymol and Carvacrol Eutectic on Growth Performance, Serum Biochemical Parameters, and Intestinal Health in Broiler Chickens"

_animals, 2023, doi:10.3390/ani13132242_

Round 1
Reviewer 1 Report
#62 add 'the' - .....which may lead to the deterioration [11]
#89 What is the sex? If you used mixed, how many male or female?
#95-96 Need to include more information on the husbandry of the birds, e.g. room temperature?
#108-109 rewrite the sentence: At 21 and 42 days of age, after 12 h of feed withdrawal, birds were weighed, and feed consumption was obtained by cages. What do you mean got by cages?
#110 How to use the weight of dead birds? May be need to reconstruct the sentence.
#117-118 Reconstruct the sentence: The jejunum and ileum segments were collected and stored in 4% paraformaldehyde solution for the preservation and then to determine intestinal morphology.
Below are few additional comments:
~This study tried to find out the effect of diets supplementing with various levels of thymol and carvacrol eutectic on growth performance, serum biochemical parameters, intestinal morphology, and the expression of intestinal nutrient absorption, barrier function- and inflammation-related genes in broiler chickens. There is the possibility to use these compounds to substitute the in-feed antimicrobial growth promoter.
~I consider the topic original and relevant in the field and it addressed the gap in the field.
~The current study used specific compounds thymol and carvacrol eutectic, which can be found in plant essential oil. Other publications used a general plant essential oil without identifying the contents of the plant oil.
~The methodology write-up needs to be improved as this is important to relate to the statistical analysis. e.g. sexes of the chicks, if use mixed sexes, we have to ensure the number of males or females. The way of statistical analysis may be different.
~The references are appropriate.
~No comments on the tables and figures until a detailed methodology which will be explained by the authors.
Need to be improved further
Sentences must be written short and precise. Meaning of sentences may mislead the readers.
Author Response
请参阅附件。

Reviewer 2 Report
The paper tries to show the potential benefits of using a TCE of carvacrol + thymol in broiler chickens’ performance and some relevant biological parameters. However, they don’t explain which are the real benefits of using a TCE formulation in front of other available technologies (coating for examples). Furthermore, they try to conclude that this technology is better than the use of natural essential oils or a simple mixture of the active compounds of the TCE without any study where they compare both formulas. In conclusion, I don't see any potential benefits of supplying active compounds by this methodology instead of the simple application of the active compounds, because there is no real comparison at all.
I have some concerns about the statistical treatment of the data. Was the zero included in the models to assess the linearity or quadratic effects? Did they apply the proper adjustments for productive parameters (Tuckey’s suggested)?
Extensive editing of English language required
Reviewer 3 Report
The aim of the study was to evaluate the the effect of diets supplementing with various levels of thymol and carvacrol eutectic (TCE) on growth performance, serum biochemical parameters, intestinal morphology, and the expression of intestinal nutrient absorption, barrier function- and inflammation-related genes in broiler chickens. The subject of the manuscript falls within the general scope of the journal. The experiment has been well planed. The Materials and Methods section is very precise. The results have been clear presented and well disscused. In my opinion, the manuscript requires only minor revisions before publication.
Lines 16-19: I suggest to remove.
Lines 39-40: „thymol and carvacrol eutectic”. I suggest saving as second keyword.
Line 96: Please insert the value of humidity and temperature.
Line 166: „(P > 0.05)”. Please move to line 165 (after "significant effects").
Table 5: Please format the table.
Lines 212-214: I suggest to remove.
Lines 279-281: „..., which suggested that pharmaceutical cocrystal technology could improve the application effectiveness of EO in poultry production’. I suggest to remove.
Round 2
Reviewer 1 Report
#165 - The word 'not' should be 'no'.
The responses given by the authors are satisfactory. However, it still needs some minor corrections for the English and word used.
Author Response
We do really appreciate the suggestions or comments from you. Thank you for your hard work!
We have revised the manuscript one by one according to your suggestions or have given further explanation/justification for subsequent edits. The answer of suggestions are as followed.
Point 1: #165 - The word 'not' should be 'no'.
Response 1: Thank you for your suggestion. We have modified it.
Best regards,
Lixuan Li & Qiufeng Zeng
09/06/2023
Reviewer 2 Report
Line 24: without a clear comparison against an EO can't be concluded. Remove and rewrite. Furthermore, thymol and carvacrol are bioactives of essential oils, not a complete EO.
Lines 32-33: which are lineal and which quadratical. PepT1 has P=0.081 and is not significant. Correct indicating only significant changes (P<0.05)
Lines 34: IL-6 has not significant lineal effect
Line 52: Carvacrol is not an essential oil
Line 185: PepT1 not significant
Line 187: significant only respect TCE at 60 and 120 mg/kg
Figure 1: add the classification letters of the graph (a, b, c..) in a place were the can be clearly seen (top-left of each graph)
Lines 189 - 191: specify days
Lines 193-194: IL-6 lineal effect not significant. Specify days.
Lines 196-197: specify days
Line 238: higher IgG not always indicates better immune response
Lines 254 - 259: PepT1 at 30 mg/kg is not significantly different from control. Furthermore, in the reported reference they compare expression with recruitment (complete different pathways)
Conclusion: a blend of thymol and carvacrol is not an essential oil, so TCE is not a new form of essential oil. According to the description provided in the paper, could be only considered a form of administering essential oil bioactives.
Dear all,
i've detected many mistakes in the writing considering the uses of singular/plurar in some verbs in the same sentence, wrong sentences order...
I please suggest a deep English revision.
Author Response
We do really appreciate the suggestions or comments from you. Thank you for your hard work!
We have revised the manuscript one by one according to your suggestions or have given further explanation/justification for subsequent edits. The answer of suggestions are as followed.
Point 1: Line 24: without a clear comparison against an EO can't be concluded. Remove and rewrite. Furthermore, thymol and carvacrol are bioactives of essential oils, not a complete EO.
Response 1: Thank you for your suggestion. We have modified it. The experimental results showed that bioactives of essential oils treated by pharmaceutical cocrystal technology can play an active role in poultry production.
Point 2: Lines 32-33: which are lineal and which quadratical. PepT1 has P=0.081 and is not significant. Correct indicating only significant changes (P<0.05). Lines 34: IL-6 has not significant lineal effect
Response 2: Thank you very much for your suggestion. We have corrected the description of the results in the abstract.
Point 3: Line 52: Carvacrol is not an essential oil.
Response 3: Thank you for your suggestion. Carvacrol are bioactives of essential oils. We have modified it.
Point 4: Line 185: PepT1 not significant. Line 187: significant only respect TCE at 60 and 120 mg/kg.
Response 4: Thank you for your suggestion. We have modified it.
Point 5: Figure 1: add the classification letters of the graph (a, b, c..) in a place were the can be clearly seen (top-left of each graph)
Response 5: Thank you for your suggestion. We have modified it.
Point 6: Lines 189 - 191: specify days. Lines 193-194: IL-6 lineal effect not significant. Specify days. Lines 196-197: specify days.
Response 6: Thank you for your suggestion. We have added time, and modified the result description of IL-6.
Point 7: Line 238: higher IgG not always indicates better immune response.
Response 7: Thanks. Through searching relevant literature, it was found that high IgG may be caused by autoimmune diseases. So we removed.
Point 8: Lines 254 - 259: PepT1 at 30 mg/kg is not significantly different from control. Furthermore, in the reported reference they compare expression with recruitment (complete different pathways)
Response 8: Thank you for your suggestion. We removed it. PepT1 mRNA expression showed a numerical increase, but no significant difference was observed, After changing the result description, we will not discuss the result.
Best regards,
Lixuan Li & Qiufeng Zeng
09/06/2023
Round 3
Reviewer 2 Report
Accept
Fine/Minor editing required